# Procedural Outcomes in Patients Treated with Percutaneous Coronary Interventions within Chronic Total Occlusions Stratified by Gender

**DOI:** 10.3390/jcm11051419

**Published:** 2022-03-04

**Authors:** Zbigniew Siudak, Leszek Bryniarski, Krzysztof Piotr Malinowski, Wojciech Wańha, Wojciech Wojakowski, Sławomir Surowiec, Robert Balan, Sławomir Januszek, Artur Pawlik, Natalia Siwiec, Krzysztof Bryniarski, Andrzej Surdacki, Jacek Legutko, Krzysztof Bartuś, Stanisław Bartuś, Rafał Januszek

**Affiliations:** 1Faculty of Medicine and Health Sciences, Jan Kochanowski University, 25-317 Kielce, Poland; zbigniew.siudak@gmail.com; 22nd Department of Cardiology and Cardiovascular Interventions, Institute of Cardiology, Jagiellonian University Medical College, 31-202 Krakow, Poland; l_bryniarski@poczta.fm (L.B.); andrzej.surdacki@uj.edu.pl (A.S.); stanislaw.bartus@uj.edu.pl (S.B.); 3Department of Bioinformatics and Telemedicine, Faculty of Medicine, Jagiellonian University Medical College, 31-530 Krakow, Poland; krzysztof.piotr.malinowski@gmail.com; 4Department of Cardiology and Structural Heart Diseases, Medical University of Silesia, 40-635 Katowice, Poland; wojciech.wanha@gmail.com (W.W.); wwojakowski@sum.edu.pl (W.W.); 5Department of Cardiology and Cardiovascular Interventions, University Hospital, 30-688 Krakow, Poland; ssurowiec@wp.pl (S.S.); arturo.pawlik@gmail.com (A.P.); 6Department of Cardiac Surgery, Klinikum Passau, 94-032 Passau, Germany; balanrobert2003@gmail.com; 7Department of Gynecology and Obstetrics, Fryderyk Chopin University Hospital No. 1, 35-055 Rzeszow, Poland; versus_25@tlen.pl (S.J.); natalia_s@opoczta.pl (N.S.); 8Department of Interventional Cardiology, Institute of Cardiology, Jagiellonian University Medical College, John Paul II Hospital, 31-202 Krakow, Poland; kbrynia@gmail.com (K.B.); jacek.legutko@kcri.org (J.L.); 9Department of Cardiovascular Surgery and Transplantology, Jagiellonian University Medical College, John Paul II Hospital, 31-202 Krakow, Poland; krzysztof.bartus@uj.edu.pl

**Keywords:** gender, percutaneous coronary intervention, periprocedural complication, procedural mortality, chronic total occlusion, procedural success

## Abstract

It has been demonstrated that gender differences are related to different procedural and long-term clinical outcomes among a general patient population treated using percutaneous coronary interventions (PCI). The objective of our analysis was to conduct assessment regarding the relationship between gender and procedural outcomes in patients treated for PCI regarding chronic total occlusions (CTO), based on a large, real-life registry. Data used to conduct the following analysis was derived from the national registry of percutaneous coronary interventions (ORPKI), upheld in co-operation with the Association of Cardiovascular Interventions (AISN) of the Polish Cardiac Society. The study involved data procured from the registry within the period from January 2014 to December 2020. All subsequent CTO procedures recorded in the registry during that period were included in the analysis. We assessed the correlation between gender and the overall rate of periprocedural complications, procedure-related mortality, and success evaluated as TIMI flow grade 3 after the procedure by univariate and multivariable modeling. At the time of conducting our investigation, there were 162 existing and active CathLabs, at which 747,033 PCI procedures were carried out during the observational period. Of those, 14,903 (1.99%) were CTO-PCI procedures, and 3726 were women (25%). The percentage share between genders did not experience any significant changes during the consecutive years observed in the current analysis. Overall periprocedural complication rate was greater among women than men (3.45% vs. 2.31%, *p* = 0.02). A comparable relationship was noted for procedural mortality (0.7% vs. 0.2%, *p* = 0.006), while procedural success occurred more often in the case of women (69.3% vs. 65.2%, *p* < 0.001). Women were found to be more frequently affected by periprocedural complications (OR = 1.553; 95%CI: 1.212–1.99, *p* < 0.001) as well as procedural success (OR = 1.294; 95%CI: 1.151–1.454, *p* < 0.001), evaluated using multivariable models. Based on the current analysis performed on all-comer patients treated using PCI in CTO, women are affected by more frequent procedural complication occurrence as well as greater procedural success compared to men.

## 1. Introduction

Gender-related differences in the management, clinical presentation as well as outcomes among patients affected by coronary artery disease (CAD) undergoing percutaneous coronary interventions (PCIs) have been demonstrated in several aspects and are widely published [1]. Additionally, research conducted by our team resulted in the publication of several studies, in which we have shown such differences among patients with CAD and peripheral artery disease, divided according to gender [2,3,4,5]. So far, various studies have also been printed regarding differences in population characteristics, periprocedural outcomes and follow-up between sexes for patients treated with PCI within CTO [6,7,8,9,10,11,12,13,14]. Both in the general patient population with CAD and those regarded to angiographic studies, including patients undergoing treatment using PCI for CTO, women remain in the minority [15,16]. The analyses published to date allow the demonstration of the occurrence of CTO in a group of patients experiencing CAD, on average, at approximately 15–25% and up to 50% in those undergoing coronary angiography, which is a significant percentage of patients who are usually older and more often have undergone coronary revascularization procedures in the past, both via percutaneous and cardiosurgical methods [17]. The prevalence of female gender in patients treated within CTO varies and usually, does not exceed 20% [7]. Despite such a high frequency of CTO patients, presently, only a small percentage of them undergo percutaneous revascularization attempts, and in recent papers, their frequency amounts to several percent 3–4%, while in the past, this was even estimated at more than 10% [18,19]. In the currently analyzed database, the incidence of PCI CTOs in recent years is estimated at 2.34%, with a tendency towards decline [20]. Most of the studies published so far have been carried out on small patient groups treated using PCI in CTOs (with invasive selection criteria), and there are no existing studies among large numbers of patients, not selected in terms of operator advancement, center volume and experience, treatment success or other aspects.

The objective of the present research was assessment of the correlation between sex and procedure-related outcomes within the all-comers group treated with PCI within CTO among a large, real-life national registry.

## 2. Methods

### 2.1. Materials

Data used to carry out the present analysis were procured from the national registry of percutaneous coronary interventions (ORPKI), with is run in co-operation with the Association of Cardiovascular Interventions (AISN) of the Polish Cardiac Society. The register does not require any fees and it covers most catheter laboratories (CathLabs) conducting PCIs in Poland. This registry has been previously characterized in other articles [20]. In our research, we have included data taken from this registry between the period of January 2014 to December 2020.

### 2.2. Definitions

CTO is defined in coronary angiography as coronary occlusion without the procedure of antegrade filling the distal vessel in a manner other than through collaterals evaluated via Thrombolysis in Myocardial Infarction at the 0-grade level. It was required that the occlusion duration be longer than 3 months, which was assessed from the time of clinical event onset, i.e., myocardial infarction (MI), abrupt onset or deterioration regarding chest symptoms, demonstrated by angiography and/or confirmed by an experienced operator. For present analysis, we considered the CTO operator as the person who performed at least one CTO procedure as first operator during the analyzed period of time.

### 2.3. Statistical Analysis 

Continuous variables are demonstrated as means [standard deviation] and median ÷ interquartile range. Normality was evaluated using the Shapiro–Wilk test. Equality of variance was assessed with Levene’s test. Between-group d were compared via the Student’s or Welch’s *t*-tests, which depended on the variance equality for normally distributed variables. Categorical variables were subjected to comparison using Pearson’s chi-squared or Fisher’s exact test in cases when 20% of cells had a projected count lower than 5 (Monte Carlo simulation for Fisher’s test using tables of dimensions higher than 2 × 2). All baseline or demographic characteristics were employed as potential predictors of complications related to the procedure, death or procedure-related success in the case of univariable logistic regression models. Variables having a *p*-value < 0.2, or variables indicating clinical significance were incorporated into the multivariable model. Final multivariable logistic regression models were built using minimization of Akaike Information Criterion to distinguish stroke predictorsin the DCA and PCI +/− DCA groups. We carried out statistical analysis with R, version 3.5.3 (R Foundation for Statistical Computing, Vienna, Austria, 2019), employing the ‘lme4′ option, version 1.1–21.

## 3. Results

At the time of the evaluated period, 162 CathLabs were active. At them, 747,033 PCI procedures were conducted at that time. Furthermore, among those, 14,903 (1.99%) were CTO-PCI procedures, including 3726 were women (25%). The percentage share between genders did not experience any significant changes (*p* = 0.19) during the consecutive years of observation in the present analysis (Figure 1).

### 3.1. General Characteristics at Baseline

The examined women were older (68.7 ± 9.2 vs. 64.6 ± 9.7 years, *p* < 0.001) and more often suffered due to arterial hypertension (*p* < 0.001), diabetes mellitus (*p* < 0.001) as well as kidney-related disease (*p* < 0.001) (Table 1).

### 3.2. Coronary Angiography and Procedural Indices

Considering vascular access, women were treated more often from femoral position and less from radial when compared to men (*p* = 0.003). Women more frequently underwent stent implantation (65.1% vs. 61%, *p* < 0.001). This difference was marked for drug-eluting (DESs) (*p* < 0.001) and bare-metal stents (BMSs) (*p* = 0.03). Furthermore, drug-eluting balloons (DEBs) were used more frequently among women than men (*p* = 0.03). Additionally, the amount of contrast used (*p* < 0.001) as well as radiation exposure (*p* < 0.001) during a single procedure was greater among men than women (Table 2). 

### 3.3. Procedure-Related Complications

Procedural mortality was greater among women than men (0.7% vs. 0.2%, *p* = 0.006), similarly as in the case of overall rate of complications (3.45% vs. 2.31%, *p* = 0.02). There were no significant gender-related differences for other particular procedural complications (Table 3). 

### 3.4. Pharmacotherapy during the Procedure

Women received the clopidogrel loading dose more frequently during PCI (*p* = 0.004) when compared to men. Pharmacotherapy details during angiography and PCI are presented in Table 4. 

### 3.5. Risk Factors for Procedural Death 

We confirmed male gender among the significant risk factors of lower frequency regarding procedural mortality among the all-comer patient group treated using PCI in CTOs by univariate analysis (OR: 0.333; 95% CI: 1.018–1.109, *p* = 0.007); however, this was not confirmed via the multivariable model (OR: 0.389; 95% CI: 0.137–7.268, *p* = 0.07) (Appendix A and Figure 2).

### 3.6. Risk Factors for Overall Periprocedural Complications

Male sex was noted as a predictor for lower frequency of procedure-related complications among the all-comer patient group treated using PCI in CTOs by univariate analysis (OR: 0.651; 95% CI: 0.512–0.826, *p* < 0.001). This was confirmed via the multivariable model (OR: 0.643; 95% CI: 0.502–0.824, *p* < 0.001) (Appendix A and Figure 3). 

### 3.7. Predictors of Procedural Success

Men were found to be a predictor for lower frequency of procedural success expressed as the frequency of patients with TIMI flow grade 3 after the procedure among the all-comer patient group treated using PCI in CTOs by univariate analysis (OR: 0.828; 95% CI: 0.764–0.896, *p* < 0.001), and this was confirmed via the multivariable model (OR: 0.772; 95% CI: 0.687–0.868c, *p* < 0.001) (Appendix A and Figure 4). 

## 4. Discussion

Considering the major findings of the presented analysis, including the all-comer patient group treated using PCI in CTOs, every 5th patient in this group was treated for CTO. Additionally, women constituted every 4th patient treated with PCI within the sub-group of patients treated via PCI in CTOs. Furthermore, this percentage did not change statistically significantly in subsequent years, i.e., within the period from 2014 to 2020. Women were also, on average, older and more burdened with comorbidities. Looking at the main endpoints, women were related to the occurrence of greater procedural complications as well as higher procedural success rate compared to men. 

An undoubted, great advantage of the presented study is the fact that it is a “real-life” registry, in which all patients undergoing PCI procedures were considered; those procedures performed by less- and more-experienced operators, independently of the center volume, final clinical outcome or the prognosis of the artery for success. Therefore, patients were not selected, namely those for whom the procedure was successful, and the patency of the artery was recreated or those who previously had fewer failures with PCI within CTO. The nature of such a register generates certain differences from the designed registers kept by excellent operators working in well-experienced and high-volume centers or randomized trials, where the selection applies only to patients undergoing PCI within CTO. This distinguishes this analysis from several works on this subject available in literary studies.

When analyzing the data from the current registry, what can first be noticed is the lower frequency of CTO procedures in the all-comer patient group treated using PCI and the tendency towards a decrease, regardless of the fact that there was high confirmation supporting the appropriateness and benefits of using this type of revascularization in patients with CAD [21,22,23,24,25,26]. One of the main factors influencing this trend is the need to demonstrate a relatively large area of ischemia within the area of the artery that is occluded, and additionally, the viability of the muscle in this area [27]. Another issue is that there is a lack of hard-evidence regarding the apparent influence on follow-up mortality rate among patients undergoing PCI within CTO [28,29,30]. The greater percentage of women in the present research, in comparison to others, can certainly be mostly explained by the lack of patient selection in this registry when compared to most published works. This may be related to worse prognosis in women undergoing PCI within CTO procedures, in terms of greater risk of periprocedural death and other complications, often more technically difficult PCIs and a lower chance of retrograde procedures due to coronary artery morphology and anatomy among women [8]. The women in our analysis more frequently underwent femoral access, which may also relate to the occurrence of vascular problems, the patient and operator approaches, as well as possibilities for performing PCI within CTO [31]. Another issue in women is the higher frequency of applying pharmacotherapy alone with CTO rather than pharmacotherapy combined with PCI [10].

One of the main indicators of a different selection of patients expressed by lesion complexity, operator variety or center choice is the achieved effectiveness of the procedure, namely its relatively low level compared to other studies [13]. In most trials, no statistical differences were shown in the effectiveness of PCI within CTO distinguished according to gender [8]. However, there are works available, in which this effectiveness was statistically significantly higher among women in comparison to men [13]. Analogously, in most studies, no differences were noted during the period of long-term, follow-up, as well as in publications where a relationship was also found, worse outcomes were reported for women and they were mainly focused on higher mortality [9,10,11,32,33]. In the present study, it was not possible to compare target lesion complexity between both sexes using sophisticated tools, such as the Japanese-CTO (J-CTO) score, in contrast to other published studies in which no differences were shown at baseline [11]. However, despite the comparable degree of coronary atherosclerosis advancement, expressed as the existence of left main coronary artery and multi-vessel diseases, procedures in men were associated with significantly more contrast and radiation exposure. It could certainly be presumed that at first, this may be the consequence of the higher average weight of men, greater diameter and length of the treated arteries as well as greater body surface area. This undoubtedly had influence on patient translucency, which increases both contrast use and radiation exposure. 

By analyzing the type of artery treated, both in the group of women and men, the coronary artery (RCA) on the right side was the most often revascularized, similarly as in previously published studies, and the frequency did not differ between the sexes [26]. The only significant difference in the present analysis remained in the anterior descending artery on the left, which was treated with PCI more often in women.

The differences in the frequency of administering the loading dose regarding the second antiplatelet drug concerning the present analysis (observed more often in women) may, in the first place, be a consequence of the higher percentage of successful procedures resulting in stent implantation among the group of women.

Summarizing the results of the presented research, it seems that women undergoing PCI of CTO lesions tend to present extreme results, more often than men. Specifically, they are, on the one hand, more frequently effective, while on the other, they are burdened with a greater frequency of fatal complications, which include periprocedural death or cardiac arrest. It is very difficult to suggest any practical advice for operators qualifying patients for revascularization procedures in CTO lesions. Intuition and experience are certainly of great importance here, because it is known in advance that women’s vessels are more fragile and smaller in diameter, while surgical complications leading to a fatal outcome are a consequence of this. They are, in part, related to different vascular access and the associated greater likelihood of more serious bleeding complications, as well as serious complications within the coronary vessels, i.e., extensive dissections, dangerous perforation that may more often lead to periprocedural infarctions affecting the hemodynamics of systemic circulation.

The analysis of causes related to higher death rate among women compared to men is much more difficult because we have raw data, without access to a patient’s disease history and angiography, because the registry is anonymized. Therefore, it can only be indirectly concluded that the cause may be a greater number of vascular complications, such as bleeding from the puncture site or perforation. Nevertheless, this reasoning is highly flawed.

## 5. Conclusions

Based on the current analysis performed on all-comer patients, compiled in a real-life registry and undergoing treatment using PCI in CTOs, being of female gender poses a greater risk of experiencing procedure-related complications as well as procedural success compared to men. 

## 6. Limitations

We did not include several factors in the current analysis, i.e., race or socioeconomic variables. Additionally, technically considered aspects of the CTO-PCI are poorly reported in the analyzed registry (e.g., type of the microcatheter, working guidewire, type of culprit lesion approach, etc.), including precise variables assessing lesion complexity, such as the J-CTO score. The significant limitations of the conducted research are the lack of a follow-up period, which would include all complications occurring at the time of hospitalization and short- and long-term follow-up following hospital discharge. We possess only periprocedural data, which are supplemented by operators and do not include post-procedural indices in the following days of hospitalization or after discharge. Additionally, the analyzed data did not cover information concerning the type of collateral access in patients treated from dual access. We were also unable to use the J-CTO score as a predictor because of the lack of data for its individual components and the consequent inability for its calculation.

## Figures and Tables

**Figure 1 jcm-11-01419-f001:**
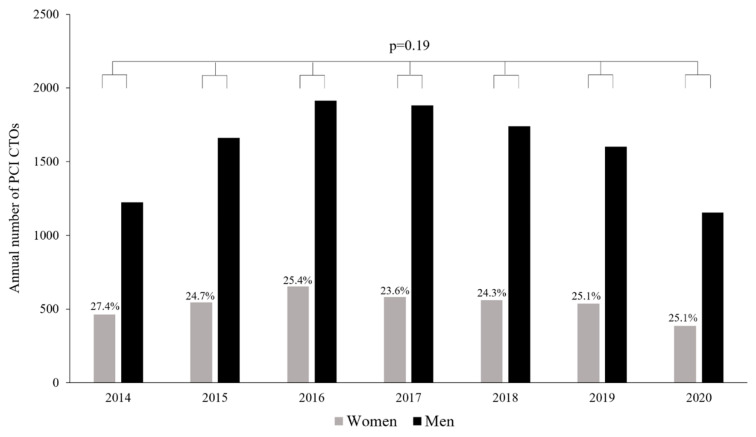
Percentage share of women and men in following years covered in the presented analysis.

**Figure 2 jcm-11-01419-f002:**
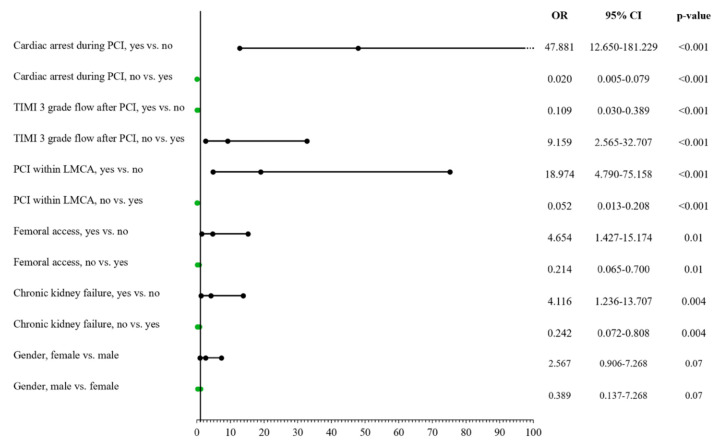
Risk factors for periprocedural death. LMCA: left main coronary artery; PCI: percutaneous coronary intervention; TIMI: Thrombolysis in Myocardial Infarction.

**Figure 3 jcm-11-01419-f003:**
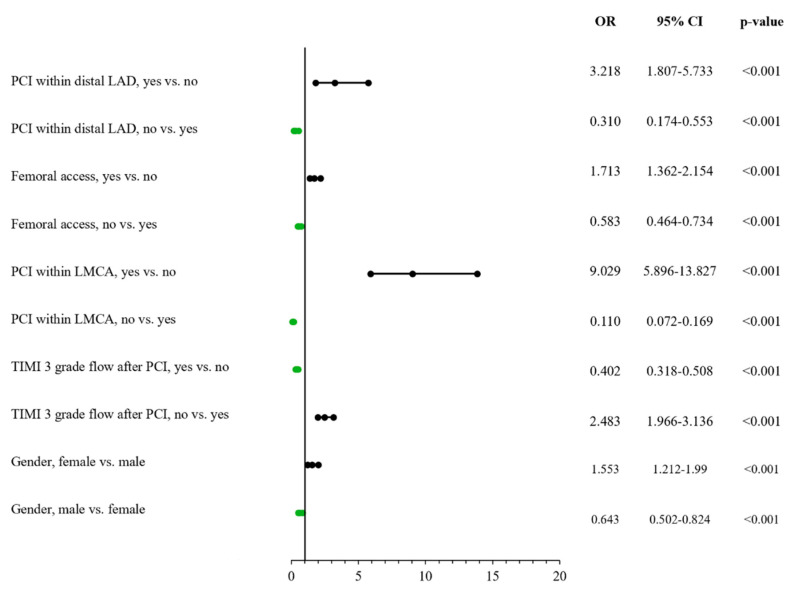
Risk factors for periprocedural complications. LAD: left anterior descendent artery; LMCA: left main coronary artery; PCI: percutaneous coronary intervention; TIMI: Thrombolysis I = in Myocardial Infarction.

**Figure 4 jcm-11-01419-f004:**
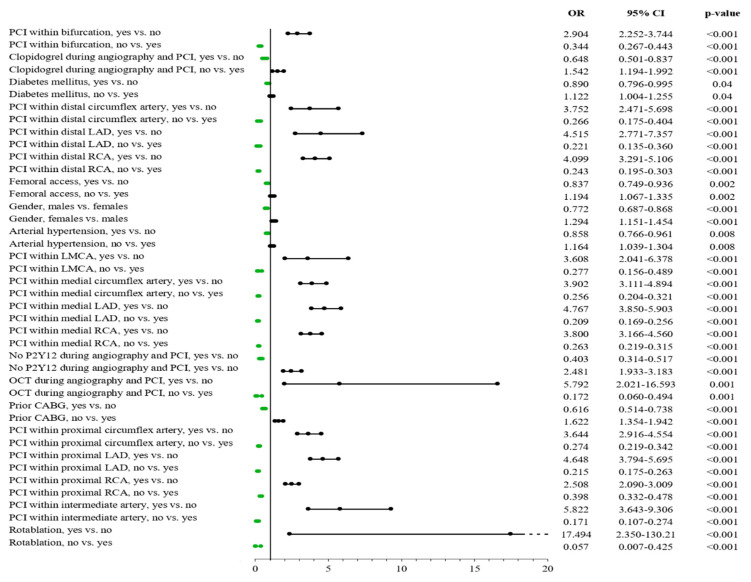
Predictors of procedural success assessed as TIMI flow grade 3 after PCI. LAD: left anterior descending artery; LMCA: left main coronary artery; OCT: optical coherence tomography; PCI: percutaneous coronary intervention; RCA: right coronary artery.

**Table 1 jcm-11-01419-t001:** Patient characteristics.

Clinical Variables	Overall (%) N = 14,903 (100)	Men (%) N = 11,177 (75)	Women (%) N = 3726 (25)	*p*-Value
Age, years	65.6 ± 9.8	64.6 ± 9.7	68.7 ± 9.2	<0.001
	66 (59 ÷ 72)	65 (58 ÷ 71)	68 (63 ÷ 75)	
Weight, kg	83.4 ± 15.8	86.2 ± 15.1	75 ± 15	<0.001
	82 (74 ÷ 92)	85 (77.5 ÷ 95)	75 (65 ÷ 83)	
Arterial hypertension	11,045 (73.9)	8153 (72.9)	2871 (77)	<0.001
Diabetes mellitus	3873 (25.9)	2662 (23.8)	1203 (32.3)	<0.001
Prior stroke	448 (3)	326 (2.9)	121 (3.2)	0.3
Prior MI	7455 (49.9)	5718 (51.2)	1720 (46.2)	<0.001
Prior PCI	8409 (56.3)	6435 (57.6)	1956 (52.5)	<0.001
Prior CABG	1213 (8.1)	974 (8.7)	238 (6.4)	<0.001
Smoking	2654 (17.8)	2169 (19.4)	478 (12.8)	<0.001
Psoriasis	54 (0.36)	42 (0.38)	12 (0.32)	0.63
Kidney disease	900 (6)	631 (5.6)	267 (7.2)	<0.001
COPD	354 (2.4)	277 (2.5)	75 (2)	0.1

Data are expressed as mean ± standard deviation and median ÷ interquartile range or numbers (percentages). CABG, coronary artery bypass grafting; COPD, chronic obstructive pulmonary disease; MI, myocardial infarction; PCI, percutaneous coronary intervention.

**Table 2 jcm-11-01419-t002:** Coronary angiography and procedural indices.

Variable	Overall (%) N = 14,903 (100)	Men (%) N = 11,177 (75)	Women (%) N = 3726 (25)	*p*-Value
Vascular access				
Femoral	2093 (24.5)	1527 (24.2)	563 (25.5)	0.003
Radial	6383 (74.6)	4752 (75.1)	1613 (73.1)	
Other	78 (0.9)	46 (0.7)	32 (1.4)	
Location of culprit lesion				
RCA	6231 (41.7)	4617 (41.3)	1600 (42.9)	0.08
LMCA	209 (1.4)	146 (1.3)	63 (1.7)	0.08
LAD	4034 (27)	2958 (26.5)	1068 (28.7)	0.009
Cx	2600 (17.4)	1981 (17.7)	610 (16.4)	0.06
Others	1868 (12.5)	1475 (13.2)	385 (11.5)	<0.001
Coronary angiography				
- SVD	12,878 (86.2)	9631 (86.2)	3213 (86.2)	0.09
- MVD − LMCA	903 (6)	690 (6.2)	211 (5.7)	
- MVD + LMCA	151 (1)	100 (0.9)	51 (1.4)	
- Separate LMCA	58 (0.39)	46 (0.4)	12 (0.3)	
- Others	952 (6.4)	710 (6.3)	239 (6.4)	
TIMI grade after PCI				
- 0	4253 (28.5)	3307 (29.7)	940 (25.3)	<0.001
- I	373 (2.5)	276 (2.5)	96 (2.6)	
- II	404 (2.7)	300 (2.7)	104 (2.8)	
- III	9869 (66.2)	7261 (65.2)	2576 (69.3)	
Rotablation	146 (0.98)	115 (1.03)	31 (0.83)	0.29
Bifurcation	1134 (7.6)	861 (7.7)	267 (7.2)	0.28
FFR	150 (1.7)	110 (1.7)	40 (1.8)	0.82
IVUS	66 (0.8)	46 (0.7)	19 (0.9)	0.53
OCT	15 (0.2)	9 (0.1)	6 (0.3)	0.23
Type of PCI				
Drug-eluting stent	9006 (60.3)	6631 (59.3)	2352 (63.1)	<0.001
Bare-metal stent	200 (1.3)	137 (1.2)	63 (1.7)	0.03
BRS	97 (0.6)	75 (0.7)	22 (0.6)	0.6
Implanted stent	9265 (62)	6818 (61)	2424 (65.1)	<0.001
Number of implanted stents				<0.001
0	5677 (38)	4359 (39)	1302 (34.9)	
1	6950 (46.5)	5096 (45.6)	1837 (49.3)	
2	1756 (11.7)	1314 (11.8)	437 (11.7)	
3	498 (3.3)	359 (3.2)	139 (3.7)	
4	56 (0.4)	44 (0.4)	11 (0.3)	
5	4 (0.03)	4 (0.04)	0 (0)	
6	1 (0.01)	1 (0.01)	0 (0)	
Number of implanted stents ≥2	2315 (15.5)	1722 (15.4)	587 (15.7)	0.6
Stent type				<0.001
BMS alone	176 (1.2)	122 (1.1)	54 (1.4)	
BVS alone	82 (0.55)	64 (0.57)	18 (0.48)	
BVS + BMS	1 (0.01)	1 (0.01)	0 (0)	
DES	8969 (60)	6607 (59.1)	2339 (62.8)	
DES + BMS	23 (0.15)	14 (0.13)	9 (0.24)	
DES + BVS	14 (0.09)	10 (0.09)	14 (0.09)	
No stent used	5677 (38)	4359 (39)	1302 (34.9)	
DEB	208 (1.7)	142 (1.5)	65 (2.1)	0.03
Contrast dose,	205.3 ± 102	209.4 ± 103.2	193.1 ± 97.6	<0.001
mL	190 (140; 250)	200 (150; 250)	175 (130; 240)	
Radiation exposure, Gy	1.53 ± 1.29	1.62 ± 1.33	1.24 ± 1.12	<0.001
	1.16	1.26	0.9	
	(0.65; 2.01)	(0.71; 2.14)	(0.5;1.62)	

Data are expressed as mean ± standard deviation and median ÷ interquartile range or numbers (percentages). BRS, bioresorbable scaffold; CTO, chronic total occlusion; Cx, circumflex artery, FFR, fractional flow-reserve; IM, intermediate artery; IVUS, intravascular ultrasound; LAD, left anterior descending artery; LMCA, left main coronary artery; MVD, multi-vessel disease; OCT, optical coherence tomography; PCI, percutaneous coronary intervention; RCA, right coronary artery; SVD, single-vessel disease; SvG, saphenous vein graft; TIMI, Thrombolysis in Myocardial Infarction.

**Table 3 jcm-11-01419-t003:** Procedure-related complications.

Type of Complication	Overall (%) N = 14,903 (100)	Men (%) N = 11,177 (75)	Women (%) N = 3726 (25)	*p*-Value
All complications	138 (2.67)	82 (2.31)	56 (3.45)	0.02
Death	17 (0.3)	6 (0.2)	11 (0.7)	0.006
MI	22 (0.4)	13 (0.4)	9 (0.6)	0.36
No-reflow	36 (0.7)	25 (0.7)	11 (0.7)	1.00
Puncture-site bleeding	11 (0.2)	5 (0.1)	6 (0.3)	0.11
Cardiac arrest	27 (0.5)	13 (0.4)	14 (0.9)	0.04
Allergic reaction	3 (0.1)	1 (0.0)	2 (0.1)	0.23
CAP	45 (0.9)	28 (0.8)	17 (1.1)	0.34
Stroke	0 (0)	0 (0)	0 (0)	-
CAD	3 (0.2)	3 (0.2)	0 (0)	0.56

Data are expressed as numbers (percentages). The χ^2^ test was used for categorical variables. CAD, coronary artery dissection; CAP, coronary artery perforation; MI, myocardial infarction.

**Table 4 jcm-11-01419-t004:** Pharmacotherapy during angiography and PCI.

Type of Complication	Overall (%) *n* = 14,903 (100)	Men (%) *n* = 11,177 (75)	Women (%) *n* = 3726 (25)	*p*-Value
Acetyl-salicylic acid	4563 (43.4)	3349 (42.9)	1193 (44.5)	0.16
Clopidogrel	5078 (48.5)	3697 (47.6)	1361 (50.8)	0.004
Prasugrel	48 (0.6)	38 (0.6)	10 (0.4)	0.42
Ticagrelor	622 (7.1)	468 (7.2)	152 (6.8)	0.47
No P2Y_12_ during procedure	4910 (46.1)	3715 (46.9)	1190 (43.9)	0.006
Unfractionated heparin	12,931 (93.3)	9697 (93.5)	3198 (92.8)	0.18
LMWH	696 (7.7)	511 (7.7)	185 (7.9)	0.68
GP IIb/IIIa	203 (1.4)	158 (1.4)	45 (1.2)	0.34
Bivalirudin	13 (0.1)	9 (0.1)	4 (0.2)	0.75

Data are expressed as numbers (percentages). The χ^2^ test was used for categorical variables. GP: glycoprotein; LMWH: low-molecular weight heparin; PCI: percutaneous coronary intervention.

## Data Availability

Data are available on reasoned request.

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
