# Peer review of "Procedural Outcomes in Patients Treated with Percutaneous Coronary Interventions within Chronic Total Occlusions Stratified by Gender"

_jcm, 2022, doi:10.3390/jcm11051419_

Round 1

Reviewer 1 Report

Congratulations, the article is interesting and the population of the study is large, even if I think it needs a little improvement.

In real life in Poland women who undergo CTO PCI are older and have more comorbidities than men, undergo more periprocedural complication/death but CTO PCI are more often successful.

I think you should explain better why if the procedures are more successful there is a higher risk of death and periprocedural complication for women.

Also, I would like to know which kind of complications occurred; for example, were they related to femoral access (which is more frequent in women compared to men) or else? Do you think that complications are gender related or that there may be a bias due to age and comorbidities?

Why do you think that success of CTO PCI was higher in women?

Figure 1 is not clear: the histogram height does not match the percentage shown. Please clarify.

Finally, there are minor English mistakes and typos throughout the document (for example line 122).

Author Response

Reviewer 1

Congratulations, the article is interesting and the population of the study is large, even if I think it needs a little improvement. In real life in Poland women who undergo CTO PCI are older and have more comorbidities than men, undergo more periprocedural complication/death but CTO PCI are more often successful.

I think you should explain better why if the procedures are more successful there is a higher risk of death and periprocedural complication for women.

This information was added to the last part of the ‘Discussion’ section.

Also, I would like to know which kind of complications occurred; for example, were they related to femoral access (which is more frequent in women compared to men) or else? Do you think that complications are gender related or that there may be a bias due to age and comorbidities?

This is presented in Table 3 and discussed in the following sections of the ‘Discussion’.

Why do you think that success of CTO PCI was higher in women?

This has been added to the last paragraph of the ‘Discussion’ section.

Figure 1 is not clear: the histogram height does not match the percentage shown. Please clarify.

The values do match.

Women

Men

Overall

%Women

2014

463

1225

1688

0.274289

2015

545

1662

2207

0.246942

2016

653

1913

2566

0.254482

2017

581

1881

2462

0.235987

2018

560

1740

2300

0.243478

2019

537

1602

2139

0.251052

2020

387

1154

1541

0.251136

Finally, there are minor English mistakes and typos throughout the document (for example line 122).

These minor errors have been corrected.

Reviewer 2 Report

The article presents the analysis of periprocedural complications and efficacy of planned PCI for chronic coronary artery occlusion depending on the gender of the patient.

Given the increase in the prevalence of coronary artery disease, the increase in interventional activity in the management of patients with stable angina pectoris, the topic of the article seems to be relevant.

The article is well structured

The results presented in the manuscript are reproduced based on the verified and validated research methods described in the corresponding section. Methods of statistical analysis correspond to the purposes of the study.

Figures and tables display correctly the results obtained and are easy to understand and interpret.

In the Introduction and Discussion, the authors refer to relevant literature sources.

Conclusions reflect the results obtained.

The authors correctly and exhaustively set out the limitations of the study.

Remarks: in the Discussion, I would like authors to add the supposed practical application of the results obtained.

Author Response

Reviewer 2

The article presents the analysis of periprocedural complications and efficacy of planned PCI for chronic coronary artery occlusion depending on the gender of the patient. Given the increase in the prevalence of coronary artery disease, the increase in interventional activity in the management of patients with stable angina pectoris, the topic of the article seems to be relevant. The article is well structured. The results presented in the manuscript are reproduced based on the verified and validated research methods described in the corresponding section. Methods of statistical analysis correspond to the purposes of the study. Figures and tables display correctly the results obtained and are easy to understand and interpret. In the Introduction and Discussion, the authors refer to relevant literature sources. Conclusions reflect the results obtained. The authors correctly and exhaustively set out the limitations of the study.

Thank you very much for the very positive assessment of the manuscript presented for review.

Remarks: in the Discussion, I would like authors to add the supposed practical application of the results obtained.

This information has been added to the last paragraph of the ‘Discussion’ section.

Reviewer 3 Report

Manuscript: Procedural Outcomes in Patients Treated with Percutaneous Coronary Interventions Within Chronic Total Occlusions Stratified by Gender

The authors investigated the relationship between gender and procedural outcomes among patients treated with PCI within chronic total occlusion (CTO). The study covered data obtained from the registry from January 2014 to December 2020. All subsequent CTO procedures recorded in the registry during that period were included into the analysis. The authors assessed the relationship between gender and overall periprocedural complication rate, procedural mortality and procedural success. During the investigated period, 747,033 PCI procedures were performed. Of those, 14,903 (1.99%) were CTO-PCI procedures, and 3,726 were women (25%). Overall periprocedural complication rate was greater among women than men (3.45% vs. 2.31%, p=0.02). A similar relationship was observed for procedural mortality (0.7% vs. 0.2%, p=0.006), while procedural success occurred more often among women (69.3% vs. 65.2%, p<0.001). Women were found to be more often related to periprocedural complications (OR=1.553; 95%CI: 1.212-1.99, p<0.001) and procedural success (OR=1.294; 95%CI: 1.151-1.454, p<0.001), evaluated using multivariable models. The authors concluded that in patients treated with PCI within CTO, women are related to greater procedural complication occurrence as well as greater procedural success compared to men.

In my opinion, this MS has three serious and mayor limitations:

  • Gender differences in CTO PCI have been reported widely in the literature. The authors should state clearly in the text which are, under their point of view, the original contributions to the field.
  • Technical variables are poorly reported.
  • Lack of follow up (as acknowledge by the authors in the limitations)

Author Response

Manuscript: Procedural Outcomes in Patients Treated with Percutaneous Coronary Interventions Within Chronic Total Occlusions Stratified by Gender

The authors investigated the relationship between gender and procedural outcomes among patients treated with PCI within chronic total occlusion (CTO). The study covered data obtained from the registry from January 2014 to December 2020. All subsequent CTO procedures recorded in the registry during that period were included into the analysis. The authors assessed the relationship between gender and overall periprocedural complication rate, procedural mortality and procedural success. During the investigated period, 747,033 PCI procedures were performed. Of those, 14,903 (1.99%) were CTO-PCI procedures, and 3,726 were women (25%). Overall periprocedural complication rate was greater among women than men (3.45% vs. 2.31%, p=0.02). A similar relationship was observed for procedural mortality (0.7% vs. 0.2%, p=0.006), while procedural success occurred more often among women (69.3% vs. 65.2%, p<0.001). Women were found to be more often related to periprocedural complications (OR=1.553; 95%CI: 1.212-1.99, p<0.001) and procedural success (OR=1.294; 95%CI: 1.151-1.454, p<0.001), evaluated using multivariable models. The authors concluded that in patients treated with PCI within CTO, women are related to greater procedural complication occurrence as well as greater procedural success compared to men.

In my opinion, this MS has three serious and mayor limitations:

  • Gender differences in CTO PCI have been reported widely in the literature. The authors should state clearly in the text which are, under their point of view, the original contributions to the field.

This is explained, inter alia, in the second paragraph of the ‘Discussion’ section.

  • Technical variables are poorly reported.

Unfortunately, the database analysed by our team (PCI register) does not contain such data although we would very much like to analyse such data because we are aware that it has impact on the final assumptions, and its absence may lead to drawing erroneous conclusions. For this reason, a relevant note has been added to the ‘Limitations’ section.

  • Lack of follow up (as acknowledge by the authors in the limitations)

We are aware of this disadvantage of the work, which is why we have included such a note in the ‘Limitations’ section. Nevertheless, an undoubted advantage is the large registry without segregating patients., as is usually the case in the vast majority of registries kept by widely-recognised PCI leaders within CTO. A wide range of operators and CathLabs have been included in this database and an analysis has been included in the ‘Discussion’ section, including tertiary centres and novice operators to leading academic centres and CTOs, world-class operators.

Round 2

Reviewer 3 Report

I thank the authors for the comments and the inclusion of my previous concerns in the limitations.